# Anticancer Potential of Isoflavones: A Narrative Overview of Mechanistic Insights and Experimental Evidence from the Past Ten Years

**DOI:** 10.3390/biomedicines13122990

**Published:** 2025-12-05

**Authors:** Maryna Schuenck Knupp, Lucas Nicolau de Queiroz, Mateus de Freitas Brito, Lucas Silva Abreu, Bruno Kaufmann Robbs

**Affiliations:** 1Postgraduate Program in Applied Science for Health Products, Faculty of Pharmacy, Fluminense Federal University (UFF), Niteroi 24241-000, RJ, Brazil; marynask@id.uff.br (M.S.K.); lucasnicolaunf@gmail.com (L.N.d.Q.); 2Natural Products Chemistry Laboratory, Institute of Chemistry, Federal Fluminense University (UFF), Niteroi 24210-201, RJ, Brazil; mateusfb@id.uff.br; 3Basic Science Department, Health Institute of Nova Friburgo, Fluminense Federal University (UFF), Nova Friburgo 28625-650, RJ, Brazil

**Keywords:** antitumor activity, cytotoxicity, isoflavones, natural products, phytoestrogens

## Abstract

Isoflavones are natural compounds abundantly found in soybeans, recognized for their anticancer potential, primarily through their activity as phytoestrogens, which inhibit estrogen receptors. Because cancer remains one of the leading causes of mortality worldwide, identifying compounds that may complement chemotherapy is of great interest. In this review, we summarize advances reported over the past decade regarding the antitumor properties of isoflavones, with emphasis on both in vitro and in vivo effects, as well as chemical, botanical, and pharmacological aspects. A literature search was conducted using the PubMed database covering studies published from January 2014 to April 2025 using the following keywords: ‘isoflavones’ and ‘anticancer’, ‘antitumoral’, and ‘antiproliferative’ and ‘cytotoxicity’. Genistein and daidzein emerge as the most extensively studied isoflavones, with well-documented anticancer activity. Reported anticancer effects include induction of apoptosis, ROS generation, cell cycle arrest, inhibition of cell migration and invasion, loss of mitochondrial membrane potential, modulation of estrogen-related pathways, and antiangiogenic activity. In addition to these mechanistic findings, several isoflavones demonstrated significant tumor growth inhibition in xenograft models, reinforcing their translational potential. Additionally, synergistic interactions with chemotherapeutic drugs and natural compounds and new drug delivery systems have been described. Breast and prostate cancer cell lines were the most investigated due to isoflavones’ estrogen-like effects. However, the cell death mechanisms of newly discovered compounds still require further investigation.

## 1. Introduction

Cancer remains one of the leading causes of mortality worldwide. In 2022, approximately 20 million new cases were reported across 185 countries [1]. Given this scenario, the ongoing search for effective treatment options is imperative. Chemotherapy remains one of the most effective approaches for inhibiting tumor cell proliferation. Therefore, continued research is essential to identify bioactive compounds that can be integrated into such treatments. Plant-derived molecules have shown remarkable anticancer activity and have been used as chemotherapeutic agents [2].

Natural products have attracted considerable interest from the pharmaceutical industry, particularly due to the advances achieved over the past 50 years. Notably, antitumor therapy has made significant progress as a result of these discoveries [3]. Consequently, a large proportion of antineoplastic therapies approved in recent decades have a direct or indirect origin in natural products, whether through isolated molecules, semisynthetic derivatives, or compounds inspired by their chemical structures [4]. The most well-known examples are the vinca alkaloids (vincristine and vinblastine) and Paclitaxel, which was isolated from the plant species *Taxus brevifolia* [5]. These compounds frequently act on key cellular signaling pathways, including microtubule formation, DNA synthesis, apoptosis, oxidative stress, the inhibition of cell cycle progression, angiogenesis and cell migration [6]. Isoflavones are a remarkable example of a class of natural products with antitumor potential.

Belonging to one of the most common categories of phytoestrogens, isoflavones are polyphenolic compounds. They are secondary metabolites produced by plants, whose structure is similar to that of the hormone 17β-estradiol. They are found mainly in the legume family (*Fabaceae*), usually in a conjugated form. In the diet of farm animals, soybean (*Glycine max*), red clover (*Trifolium pratense*) and white clover (*Trifolium repens*), and alfalfa (*Medicago sativa*) are important isoflavone sources. Humans obtain most of their dietary isoflavones from soy and soy-based products [7].

The first report of isoflavone activity in tumor cells dates to the 1970s, when cell cytotoxicity was demonstrated [8]. Around the same time, it was discovered that this class of molecules acts as phytoestrogens, showing strong interaction with estrogen receptors in breast cancer models [9]. Since then, several research groups have been working to elucidate the antineoplastic mechanisms of isoflavones through in vivo, in vitro, and in silico studies, aiming to develop drug prototypes and incorporate them effectively into chemotherapy treatments. Subsequently, the high chemopreventive potential of soy-derived isoflavones was identified, with these compounds being among the most comprehensively investigated in experimental cancer models [10].

Over time, new isoflavones have been discovered from natural sources. Recent studies have shown that, in the past ten years, approximately 1036 new prenylated flavonoids have been isolated from 127 plant species, among which 219 are isoflavones [11]. This expands the prospects for investigating their biological activities. In a recently published review, newly obtained synthetic or hybridized derivatives were shown to exhibit strong antitumor activity [12]

However, the last comprehensive review addressing the anticancer activity of isoflavones was published over a decade [13]. More recent reviews have tended to focus on specific cancer types, individual isoflavones, or the nutritional aspects of isoflavones in relation to antitumor activity and signaling pathways. In light of this, the present work aims to summarize the findings of the past decade regarding the antitumor mechanisms of action and experimental approaches of molecules from this class, whether derived from natural products or synthetic ones.

## 2. Methodology of Literature Review

### 2.1. Type of Review

This work is a narrative literature review, designed to synthesize recent experimental evidence on the anticancer properties of isoflavones. Given the wide heterogeneity of study designs, biological models, and mechanistic endpoints reported in the literature, a narrative review framework was considered the most appropriate approach for summarizing current knowledge without imposing the methodological constraints of a systematic review.

### 2.2. Search Strategy

The literature survey was conducted using the PubMed database and covered the period January 2014 to April 2025. The search strategy used the keywords “isoflavones” combined with at least one additional term (“anticancer,” “antitumoral,” “antiproliferative,” or “cytotoxicity”) appearing in the title or abstract. This strategy allowed us to identify studies specifically investigating experimentally validated anticancer activity.

### 2.3. Study Selection Process

Studies were included when they provided experimental in vitro and/or in vivo evidence of the antitumor activity of structurally elucidated isoflavones. Articles were excluded when they:Described only in silico results;Did not directly assess anticancer activity;Examined compounds whose chemical structures had not been elucidated; orWere written in languages other than English.

Because many newly identified isoflavones were reported in single, isolated publications, including highly heterogeneous methodologies, compiling them directly into the main dataset would reduce comparability and distort the identification of recurrent mechanisms. Nonetheless, to ensure transparency and avoid selection bias, all such single-reference isoflavones were cataloged and listed in the Appendix A, including their reported biological activities and corresponding references.

### 2.4. Data Analysis

For the narrative synthesis presented in the main text, we conducted a more detailed descriptive analysis only for isoflavones that had been investigated in two or more independent publications. This criterion allowed us to extract mechanisms that were recurrent or at least independently reproduced, avoiding overinterpretation based on isolated findings.

For these compounds, we summarized experimental data including:Antitumor mechanisms;Gene and protein expression modulation;Effects on cell death pathways;Cell line models used;In vivo xenograft results (when available);Overall biological outcomes in relation to the methodological approaches.

Language editing was supported using the TRINKA grammar tool; all text generated or modified through this tool was subsequently reviewed, edited, and finalized by the authors, who take full responsibility for the content.

## 3. Anticancer Activity of Isoflavones

The chemical structures and names of antitumor isoflavones are provided in Figure 1. The results and discussion were limited to substances supported by more than two independent references, as these offer more consistent and validated findings, and are summarized in Table 1. Substances with only a single reference, which typically reflect preliminary or less validated results, are summarized separately in Appendix A for completeness.

### 3.1. Genistein

Genistein is the most extensively studied soy-derived isoflavone, with 20 out of the 83 articles referenced in this review (approximately 24%) addressing its antitumor activity. Recently, it was isolated from *Pueraria lobata* roots and derivative from *Maclura cochinchinensis* (*Cudrania cochinchinensis*) [38,39]. Genistein exhibits cytotoxicity against breast adenocarcinoma cell line MCF-7 and MDA-MB-231, prostate adenocarcinoma (PC-3), colon adenocarcinoma (HT-29) and retinoblastoma cell line (Y79) [28,38,39,45,48,50]. This compound suppresses the proliferative activity of tumor cells expressing constitutively active *Src*. In gallbladder carcinoma cells transfected with the *v-src* oncogene (HAG/src3-1), genistein significantly reduces cell growth compared with the corresponding control line transfected with the pSV2neo plasmid (HAG/neo3-5). These findings demonstrate that genistein effectively inhibits the proliferation of *Src*-transformed cells under the experimental conditions evaluated, without suggesting additional mechanistic pathways beyond those directly supported by the reported data [44]. Furthermore, the complexation of genistein with phospholipids, in the form of genistein–phytosomes, inhibits cell proliferation against hepatocellular carcinoma cell line (HepG2) [40].

Several genistein derivatives also exhibit notable anticancer activity. Genistein-8-C-glucoside induces cell death in the ovarian carcinoma cell line SK-OV-3 [41]. *3′*-Hydroxygenistein shows activity in HT-29 cells, while 6,8-diprenylgenistein exhibits cytotoxic effects against the epidermoid carcinoma cell line KB and the hepatocarcinoma cell line HepG2 [39,42]. Genistein-7-glucoside (genistin) decreases cell viability in MDA-MB-231 cells overexpressing ERα (Estrogen Receptor Alpha) as well as in MCF-7 cells [43]. The thiolated genistein analog (thiogenistein) demonstrates activity against the prostate cancer cell line DU-145 [46]

The combination of genistein and equol exhibits cytotoxicity against breast adenocarcinoma cell lines (MCF-7, MDA-MB-468, and SK-BR-3) [37]. A synergistic effect between tyrosine kinase inhibitors (gefitinib, erlotinib, afatinib and AZD9291) with genistein, as well as other isoflavones (glycitin, daidzin, biochanin A, and formononetin), induces cell death through interactions with the epidermal growth factor receptor (EGFR) in non-small-cell lung cancer (NSCLC) cells [18]. The dual effect of isoflavones when combined with clinically used chemotherapeutic agents is particularly noteworthy. Their ability to interact with estrogen receptors constitutes a mechanism capable of enhancing antitumor activity, as this interaction may suppress proliferative signaling pathways that are not targeted by the accompanying drug. Moreover, the combined action of multiple isoflavones may further influence this interaction, given that individual isoflavones may exhibit differing affinities for specific estrogen receptor subtypes, thereby increasing the likelihood of an effective mechanism of action. It has also been demonstrated that isoflavones (particularly those derived from soy) exert chemopreventive properties, including anti-inflammatory and antioxidant effects in breast cancer models, while simultaneously modulating the expression of genes involved in cell cycle progression and apoptosis induction (as detailed throughout the text) [61,62]. Taken together, these findings reinforce the relevance of this dual activity.

Furthermore, different genistein formulations may also enhance its biological activity. The complex formed between genistein and hydroxypropyl-β-cyclodextrin at a 1:1 molar ratio improves cytotoxicity in cervical adenocarcinoma (HeLa), skin epidermoid carcinoma (A431), MCF-7, and ovarian carcinoma (A2780) cell lines [49]. The liver carcinoma cell line Bel-7402 exhibits reduced viability when treated with nanoparticles formulated from genistein complexed with copper and the widely used chemotherapeutic agent 5-fluorouracil (5-FU) (GEN-Cu-GEN@FUA). This effect is attributed to the ability of copper ions to deplete intracellular glutathione levels, thereby promoting oxidative stress. Notably, these nanoparticles show minimal cytotoxicity toward normal human umbilical vein endothelial cells (HUVECs) and the normal liver cell line LO2, and they demonstrate greater efficacy than either genistein or 5-FU administered individually [22,51].

On the other hand, estrogens may exert cytoprotective effects, as observed in medulloblastoma, where they can reduce sensitivity to the cytotoxic effects of cisplatin, since tumor cell growth is stimulated by 17β-estradiol. This phenomenon may contribute to increased chemoresistance. Therefore, inhibition of the estrogen receptor ERβ may enhance treatment efficacy and suppress the tumor growth induced by exogenous estradiol. Given the anti-estrogenic properties of isoflavones, they may inhibit this target, thereby acting as adjuvants to chemotherapy. Genistein exhibits a cytoprotective effect against cisplatin-induced cytotoxicity and inhibits colony formation in the medulloblastoma cell line D283 Med [24].

Apoptosis plays a pivotal role in antitumor activity by selectively eliminating abnormal or cancerous cells. This process is regulated by specific molecular cell death mechanisms that, once activated, lead to the controlled fragmentation of tumor cells. Many anticancer therapies aim to induce apoptosis, thereby restoring the balance between proliferation and cell death through intrinsic and extrinsic pathways that are frequently dysregulated in malignant tumors [63].

Genistein downregulates the expression of estrogen receptor ERα and phospho-ERα in MCF-7 cells and modulates both mitochondria-independent and mitochondria-dependent apoptotic pathways through the activation of caspases-8 and -9. Moreover, genistein treatment increases the expression of Bcl-2-associated X protein (BAX) while suppressing the expression of B-cell lymphoma 2 (BCL-2) [38]. The derivative genistein-7-glucoside likewise downregulates ERα, induces caspase-dependent apoptosis in MCF-7 and MDA-MB-231 cells, and decreases Bcl-extra-large (BCL-xL) expression in MCF-7 cells. These findings support the estrogen-like activity of genistein in breast cancer cell lines [43]. Nevertheless, low and physiologically relevant concentrations of genistein can reduce caspase activity in D283 Med cells, resulting in an estrogen-like inhibition of cisplatin-induced cytotoxicity through suppression of ERβ (estrogen receptor beta) [24].

This isoflavone induces dose-dependent apoptosis in PC-3 cells encapsulated in alginate hydrogel within a 3D culture model through a non-mitochondrial pathway, demonstrating that this culture system may enhance the effectiveness of treatment with this compound [45]. Additionally, it downregulates microRNA-155 (miR-155) and its target genes involved in apoptosis and glucose metabolism, such as STAT3 and hexokinase 2, particularly in tamoxifen-sensitive MCF-7 cells [50]. Other derivatives, including genistein-8-C-glucoside and 3′-hydroxygenistein, have also demonstrated pro-apoptotic activity, with evidence of morphological alterations in ovarian carcinoma cells (SK-OV-3) and apoptotic cell death in colorectal carcinoma cells (HT-29), respectively [41,42].

In addition to its individual effects, genistein exhibits enhanced pro-apoptotic activity when combined with other compounds or administered through specialized delivery systems, further reinforcing its therapeutic relevance. For example, its combination with equol markedly increases apoptosis in MCF-7 cells by elevating the BAX/BCL-xL ratio. Likewise, genistein formulated as phytosomes induces caspase-dependent apoptosis in HepG2 cells, whereas advanced nanostructures (such as the GEN-Cu-GEN@FUA nanoparticle) have demonstrated potent apoptotic effects in Bel-7402 cells [37,51]. These findings underscore the versatility of genistein as both a direct inducer of apoptosis and a synergistic agent when incorporated into novel therapeutic strategies.

The generation of reactive oxygen species (ROS) is closely linked to antitumor activity, as elevated ROS levels trigger oxidative stress in cancer cells. When excessively produced, ROS induce irreversible damage to DNA, proteins, and lipids, ultimately resulting in cellular dysfunction and the activation of programmed cell death pathways, particularly apoptosis [64]. Several anticancer compounds exert their therapeutic effects by promoting ROS accumulation within tumor cells.

Genistein promotes apoptosis in LNCaP and DU145 cells by increasing ROS levels through the mobilization of endogenous copper. This effect is mediated by the downregulation of Copper Transporter 1 (CTR1) and ATPase Copper Transporting Alpha (ATP7A), resulting in disrupted copper homeostasis and enhanced oxidative stress [22] ROS production is also induced in Bel-7402 cells by the GEN-Cu-GEN@FUA system [51]. Furthermore, 3-hydroxygenistein triggers ROS generation followed by apoptosis in HT-29 cells, while genistein-8-C-glucoside induces ROS production in SK-OV-3 cells [41,42]

In addition to its role in modulating apoptotic pathways, genistein also exerts antitumor effects by interfering with cell cycle regulation, wherein cell cycle arrest at specific checkpoints prevents uncontrolled proliferation and contributes to tumor suppression. By enforcing these regulatory points, numerous antitumor agents exert their therapeutic activity, effectively restricting cancer cell proliferation and limiting malignant progression [65].

At maximal physiological serum concentrations, genistein induces G0/G1 cell cycle arrest in both MCF-7 cells and the non-tumorigenic breast epithelial cell line HB4a. These findings demonstrate that, even at physiologically relevant levels, genistein maintains its chemopreventive properties, thereby challenging the hypothesis that dietary isoflavone supplementation promotes breast cancer [47]. Moreover, additional studies have shown that genistein modulates multiple molecular targets involved in cell cycle regulation, reinforcing its role as a potent regulator of tumor cell proliferation. This compound inhibits the mechanistic target of rapamycin (mTOR) and reduces cyclin E1 expression in Y79 cells [48]. Genistein may also suppress Src-driven proliferative activity in HAG/src3-1 cells by increasing the expression of cyclin-dependent kinase inhibitor 1A (p21), resulting in G2/M phase arrest [44]. This effect is also accompanied by upregulation of tumor protein 53 (p53) and downregulation of β-catenin interacting protein 1 (CTNNBIP1) gene expression [28]. 3’-Hydroxygenistein activates p53 and inhibits topoisomerase II in HT-29 cells [42]. In MCF-7 cells, genistein-7-glucoside downregulates ERα, leading to decreased cyclin D1 expression and G1 phase arrest. When combined with equol, it induces G2/M phase arrest, whereas equol alone promotes arrest in the G1 phase [37,43]. Finally, the nanoparticle formulation GEN-Cu-GEN@FUA induces cell cycle arrest at both the S and G2 phases in Bel-7402 cells [51].

Beyond its role in cell cycle regulation, the inhibition of cell migration contributes to antitumor activity by preventing cancer cells from disseminating to adjacent or distant tissues, a key step in invasion and metastasis. Antimigratory agents act by disrupting signaling pathways involved in cell motility, adhesion, and extracellular matrix remodeling [66]. The derivative genistein-7-glucoside, as well as the phospholipid-complexed formulation genistein–phytosomes, negatively regulates matrix metallopeptidase-9 (MMP-9) in MCF-7 and HepG2 cells, respectively [43]. Genistein downregulates microRNA-155 and its target genes, including STAT3 and hexokinase 2, thereby significantly reducing cell migration in tamoxifen-sensitive MCF-7 cells [50]. Copper homeostasis also plays an important role in genistein-mediated modulation of tumor cell motility. The GEN-Cu-GEN@FUA nanoparticle effectively inhibits the migration of Bel-7402 cells [51]. Furthermore, the mobilization of endogenous copper through the downregulation of *CTR1* and *ATP7A* leads to marked suppression of cell migration in LNCaP and DU145 cells [22]

Furthermore, contributing to its anticancer profile, antiangiogenic substances exert antitumor activity by inhibiting the formation of new blood vessels required for tumor growth and metabolic support through the modulation of key molecular components. Thus, antiangiogenic agents represent an effective strategy for controlling tumor progression and can be regulated through specific molecular pathways [67]. Complexation of genistein with hydroxypropyl-β-cyclodextrin improves its solubility and enhances its antiangiogenic activity by inhibiting blood vessel formation [49]. Similarly, the genistein–phytosome complex reduces the expression of vascular endothelial growth factor A (VEGFA) in HepG2 cells [40].

Xenotransplantation in immunodeficient mice provides a reliable preclinical model to evaluate the anticancer potential of new compounds, as the transplanted tumors retain key characteristics of the original tumor, including histology, chromosomal abnormalities, and surface antigen expression, allowing more accurate prediction of in vivo pharmacological activity [68]. Using this model, the nanoparticle GEN-Cu-GEN@FUA at 20 mg/kg significantly inhibited tumor growth in BALB/c nude mice bearing Bel-7402 xenografts, reducing tumor volume from approximately 1400 mm^3^ to 500 mm^3^ (~65% reduction) and tumor mass by ~2.0 g compared to controls after 16 days [51]. Similarly, genistein at 60 mg/kg decreased tumor volume from ~650 mm^3^ to 300 mm^3^ (~54% reduction) and tumor weight by half in Y79 xenografts, 60 days post-injection [48]. These findings demonstrate the strong in vivo antitumor effects of both agents and underscore the relevance of xenograft models for translational oncology research.

The genotoxic effect involves DNA damage, which can initiate carcinogenic processes. At physiological concentrations, the minerals zinc (Zn), calcium (Ca), and selenium (Se) significantly modulate the antigenotoxic activity of genistein against both metabolically activated and direct mutagens, 4-nitroquinoline oxide (4NQO) and 2-aminoanthracene (2AA), in LNCaP cancer cells [23]. These findings suggest that the chemopreventive potential of dietary isoflavones may be influenced by mineral balance and overall nutritional status.

### 3.2. Daidzein

Daidzein is an isoflavone that has been studied for a long time with its first appearance in PubMed in 1969 [69]. Like other isoflavones, it exhibits a high level of cytotoxicity against tumor cell lines as in MCF-7 and MDA-MB-231 cells, lung carcinoma cells (A549), HT-29 cells [25,26,27,28].

This isoflavone acts synergistically with other isoflavones and bioactive compounds. Daidzein decreases the sensitivity of D283 Med cells to cisplatin through estrogen-like effects by blocking the ERβ receptor, thereby reducing cisplatin-induced cytotoxicity and colony formation [24]. In contrast, daidzin, a daidzein derivative (daidzein-7-O-glucoside), enhances chemosensitivity to cisplatin by increasing cytotoxicity and inhibiting colony formation in osteosarcoma cell lines (U2OS, HOS, MNNG/HOS, SJSA-1, 143B, MG63, U2OS/MTX300), and may serve as a potential adjuvant to improve osteosarcoma treatment [29]. Daidzein combined with puerarin effectively suppresses cell proliferation by targeting the STAT3 and focal adhesion kinase (FAK) signaling pathway (STAT3/FAK), inhibiting the phosphorylation of both proteins in the endocervical adenocarcinoma cell line (BGC-823), showing selectivity toward the gastric epithelial cell line (GES-1), and reducing colony formation [30]. This isoflavone also suppresses prostate cancer cell growth by mobilizing endogenous cooper in LNCaP and DU145 cells [22]

This isoflavone exhibits pro-apoptotic properties. It induces apoptosis in breast adenocarcinoma cells through a caspase-dependent pathway, accompanied by a reduction in the ERα/ERβ ratio, upregulation of BAX, and downregulation of BCL-2 gene expression in MCF-7 cells [25]. Furthermore, daidzein and puerarin act synergistically, leading to BCL-2 downregulation and activation of effector caspases, thereby promoting apoptosis in BGC-823 cells [30]. Daidzin induces caspase-dependent apoptosis in 143B, SJSA-1, and U2OS cells and increases the BAX/BCL-xL ratio, an effect that is further enhanced by cisplatin sensitivity [29].

Daidzein induces oxidative stress through the generation of reactive oxygen species (ROS) in MCF-7 cells. Furthermore, daidzein decreases ERα expression while increasing ERβ receptor levels, thereby promoting ROS production [25]. Daidzein and puerarin, when acting together also induce ROS in BGC-823 cells [30]. The copper pathway is modulated by daidzein, as the copper-transporting genes *CTR1* and *ATP7A* are downregulated by daidzein signaling, resulting in an oxidative burst that leads to apoptosis in LNCaP and DU145 cells through the suppression of endogenous copper [22]. ROS plays a key role in triggering apoptosis and functions as an upstream signaling mediator, initiating cell death through a mechanism involving daidzein-mediated redox cycling of endogenous copper in MDA-MB-231 cells [27].

Daidzein suppresses cancer cell migration and inhibits the invasion of A549 cells [26]. In BGC-823 cells, it blocks migration by targeting the FAK/STAT3 signaling pathway when combined with puerarin and downregulates matrix metalloproteinase-2 (MMP-2) [30]. It also mobilizes endogenous copper through the inhibition of *CTR1* and *ATP7A* gene expression, further contributing to its anti-migratory effects. Daidzin and cisplatin, acting synergistically, suppress cell migration and invasion in 143B and SJSA-1 cells, downregulating MMP-2 and MMP-9 and reducing β-catenin protein stability, thereby blocking the Wnt signaling pathway in 143B, SJSA-1, and U2OS cells [22,29].

Other antitumor activities have been reported for daidzein, including cell cycle arrest at the G0/G1 phase and downregulation of cyclin D1 when combined with puerarin. It also induces alterations in mitochondrial membrane potential (MMP) in BGC-823 cells [30]. In xenograft mouse models (BALB/c nude mice), combination treatment with daidzin (20 mg/kg) and cisplatin (5 mg/kg) resulted in a ~90% reduction in tumor volume compared to the control group (550 mm^3^) and the treated group (50 mm^3^), 27 days after injection of the 143B cell line [29]. Under hypoxic conditions that mimic the tumor microenvironment, daidzein exhibits DNA-breaking activity through a nuclear, copper-dependent pathway [27]. Additionally, the minerals Zn, Ca, and Se modulate the antigenotoxic effects of daidzein against metabolically activated and direct mutagen 2AA, leading to the inhibition of LNCaP cell growth [23].

### 3.3. Equol

Equol, specifically (S)-equol, are products of daidzein degradation by the gut microbiota [70]. This enantiomeric form shows cytotoxicity against prostate cancer cell lines PC-3, DU145 and LNCaP [36]. Also, it induces cell cycle arrest at G2/M phase in PC-3 cells by downregulating cyclin B1 and cyclin-dependent kinase 1 (CDK1), while upregulating cyclin-dependent kinase inhibitors, i.e., cyclin-dependent kinase inhibitor 1A (p21) and cyclin-dependent kinase inhibitor 1B (p27) [36]. Apoptosis is promoted through increased expression of surface death receptor-Fas ligand (Fas-L) and the pro-apoptotic protein Bim. Additionally, this compound modulates the protein kinase B/Forkhead Box O3a (Akt/FOXO3a) signaling axis by elevating FOXO3a levels, reducing phosphorylated FOXO3a, and enhancing its nuclear stability [36]. This molecule also inhibits the growth of PC-3 xenograft tumors in BALB/c nude mice, demonstrating promising pharmacological activity. After 33 days of tumor cell injection, compared to the control group with an initial volume of 900 mm^3^, administration of 20 g/kg resulted in an approximately 50% reduction, reaching a volume of about 450 mm^3^. At the concentration of 10 g/kg, this reduction was even greater, about 60%, with a final volume of approximately 350 mm^3^, the lower dosage proved to be more effective [36].

Low physiological levels of (S)-equol can reduce caspase activity in D283 Med cells, leading to an estrogen-like inhibition of cisplatin’s cytotoxic effects [24]. In addition, minerals at physiological concentrations modulate the antigenotoxic effects of (S)-equol against both metabolically activated and direct genotoxins, such as 2AA, inhibiting LNCaP cell growth similarly to daidzein and genistein [23].

### 3.4. Wighteone

Wighteone and its isomers exhibit anticancer activity and were recently isolated from *Ficus altissima*, *Genista monspessulana* and *Ficus hispida* plant species [14,16,59,60]. Wighteone typically carries a prenyl substituent at the C-6 position of the A-ring, whereas lupiwighteone features the prenyl group at C-8, making it a positional isomer. Isowighteone, as the name suggests, is an isomer of wighteone, differing by shifts in the position of hydroxyl and/or prenyl groups within the aromatic rings [14,16,59,60].

This compound exhibits strong antiproliferative activity in the leukemia cell line (K562) and the gastric tumor cell line (AGS) [16]. Wighteone shows selectivity toward PC-3 cells over fibroblasts (normal cells), while remaining inactive against lung carcinoma cells. Isowighteone hydrate displays cytotoxicity against several cancer cell lines, including HepG2, promyelocytic leukemia (HL-60), A549, HeLa, KB, and HT-29 cells [14,59]. Lupiwighteone is highly cytotoxic to DU-145 cells while exhibiting low activity against HUVECs, demonstrating selectivity for the cancer cell line [60].

Isowighteone triggers ROS generation in HL-60 cells, and lupiwighteone in DU-145 cells. Both compounds exert pro-apoptotic effects, with the former acting via a caspase-dependent pathway [14,60]. Lupiwighteone also induces cell cycle arrest at the G2/M phase through inhibition of CDK1 and CDK2. Furthermore, it induces loss of mitochondrial membrane potential and downregulates the phosphorylated-Protein Kinase B/Total Protein Kinase B (p-Akt/Akt) ratio, as well as Vascular Endothelial Growth Factor (VEGF) expression. It also suppresses migration in DU-145 cells [14,60].

### 3.5. Durmillone

Durmillone exhibits anticancer activity and was recently isolated from the plant species Lonchocarpus bussei, Lonchocarpus eriocalyx, Millettia dura, and Millettia pachyloba [33,34,35]. It demonstrates antiproliferative activity against Breast Cancer Resistance Protein-expressing MDA-MB-231 cells (MB-231/BCRP), doxorubicin-resistant leukemia cells (CEM/ADR5000), and glioblastoma cells expressing EGFRvIII (U87MG.ΔEGFR) [34]. Durmillone shows high antiproliferative activity and selectivity toward A549, HeLa, and MCF-7 cells compared to normal bronchial epithelial cells (BEAS-2B) and HUVECs, respectively [33,35]. This isoflavone also upregulates the expression of autophagy-related genes, including microtubule-associated protein 1 light chain 3 beta (LC3-II), Beclin1, and autophagy-related protein 7 (Atg7), in HeLa and MCF-7 cells [33].

### 3.6. Neobavaisoflavone

Neobavaisoflavone demonstrated anticancer potential when used in combination with another chemotherapeutics. The combination with one of the two topoisomerase inhibitors (doxorubicin or etoposide) resulted in antiproliferative activity in anaplastic astrocytoma cells (SW1783) and glioblastoma cells (U-87), showing selectivity due to low cytotoxicity in normal astrocytes (NHA cells) [54,55]. Both cancer cell lines exhibited apoptosis and loss of mitochondrial membrane potential. Additionally, cell cycle arrest was induced at the G1/G0 phase in U-87 cells and at the G1/S phase in SW1783 cells [54,55].

### 3.7. Biochanin A

Biochanin A exhibits well-documented antitumor activity, particularly through synergistic interactions with chemotherapeutic agents. In breast cancer cells (MCF-7 and MDA-MB-231), its combination with 5-fluorouracil (5-FU) enhances cytotoxicity by inducing caspase-dependent apoptosis, reducing ER-α protein levels, and suppressing the ER-α/Akt signaling axis, thereby potentiating the pro-apoptotic effects of 5-FU [19].

Similarly, in pancreatic adenocarcinoma cells (AsPC-1 and MIA PaCa-2), Biochanin A in combination with atorvastatin promotes cytotoxicity through late apoptosis, characterized by decreased expression of phospho-Akt and phospho-mTOR, cleavage of poly (ADP-ribose) polymerase (PARP), inhibition of cell cycle progression via downregulation of Cyclins A, B1, and D1, and suppression of STAT3 activation. Additionally, co-treatment with 5-FU reduces MMP-2 expression, contributing to the inhibition of cell migration in breast cancer cell lines [20]. This compound also binds to the aryl hydrocarbon receptor (AhR), inducing the expression of anticancer-related genes in mouse colonocyte cells, including cytochrome P450 family 1 subfamily A member 1 (CYP1A1), cytochrome P450 family 1 subfamily B member 1 (CYP1B1), and UDP-glucuronosyltransferase family 1 member A1 (UGT1A1) [21].

Another important aspect of this isoflavone is its translational relevance, as in vivo studies demonstrate significant tumor growth inhibition. In a mouse model of Ehrlich solid carcinoma, tumor volume decreased by 75% compared to the mock-treated control group when mice were administered 20 mg/kg of 5-FU as a single agent, or 5 mg/kg in combination with Biochanin A, accompanied by extensive tumor necrosis, as confirmed by histological analysis [19].

### 3.8. Isolupalbigenin

Isolupalbigenin, recently isolated from *Ficus altissima* and *Erythrina poeppigiana* plant extracts, have antitumor activity [16,52,53]. This isoflavone exhibits antiproliferative activity against several tumor cell lines, including leukemia cells (K562), Adriamycin-resistant leukemia cells (K562/ADR), A549 cells, lung carcinoma cells (SPC-A1), gastric adenocarcinoma cells (AGS), HeLa cells, and nasopharyngeal carcinoma cells (CNE2). K562 cells are the most sensitive and demonstrate high selectivity compared to normal epidermal cells (HaCaT) [16]. Also, in HepG2 cell lines, induces inhibition of colony formation [52,53]. Isolupalbigenin also exerts pro-apoptotic effects in HepG2 cells, triggers caspase-dependent apoptosis in K562 and HL-60 cells and inhibits glyoxalase I (GLO I) [16,52,53].

### 3.9. Derriscandenon

Derriscandenon has also demonstrated anticancer activity and was recently isolated from the plant species *Derris scandens* [31,32]. Derriscandenon B and C exhibit antiproliferative activity and selectivity toward KB cells and the acute lymphoid leukemia B cell line (NALM-6), while showing low cytotoxicity in fibroblasts [31]. Derriscandenon E and F display antiproliferative activity and high selectivity against A549, KB, and NALM-6 cancer cell lines compared to normal dermal fibroblasts. These isoflavones also induce loss of mitochondrial membrane potential in A549 and KB cells [32].

### 3.10. Alpinumisoflavone

Alpinumisoflavone demonstrates anticancer activity and was recently isolated from *Genista monspessulana*, *Mappianthus iodoides*, and *Ficus altissima* [14,15,16]. It induces cell death and shows selectivity toward PC-3 cells and cervical carcinoma cells (SiHa) compared to mouse fibroblasts (L929), while remaining inactive against lung carcinoma cells. It exhibits cytotoxic effects against HL-60 cells, hepatocarcinoma cells (SMMC-7721), MCF-7 cells, colorectal adenocarcinoma cells (SW480), and leukemia cells (K562 and K562/ADR) [14,15,16]. Alpinumisoflavone also acts synergistically with gemcitabine, demonstrating antiproliferative activity in pancreatic cancer cells (PANC-1 and MIA PaCa-2). This synergism triggers ROS-induced apoptosis, inhibition of cell migration, cell cycle arrest in the sub-G1 phase, and loss of mitochondrial membrane potential. Furthermore, it regulates p53 and ribonucleotide reductase catalytic subunit M1 (RRM1) gene expression in PANC-1 and MIA PaCa-2 cells, which is relevant since gemcitabine resistance is associated with increased RRM1 expression, contributing to enhanced tumor cell proliferation [17].

### 3.11. Puerarin

Puerarin has been widely reported to exhibit anticancer activity. This isoflavone displays cytotoxic effects against MCF-7, MDA-MB-231, U87MG, and HeLa cells [56,57,58]. In HeLa cells, puerarin induces caspase-dependent apoptosis, whereas in U87MG cells it enhances ferroptosis sensitivity through autophagy regulation mediated by sirtuin 3 and nuclear receptor coactivator 4 (SIRT3/NCOA4) [57,58]. These effects are characterized by increased cytotoxicity, elevated levels of ROS and intracellular Fe^2+^, a reduced ratio of reduced to oxidized glutathione (GSH/GSSG), and upregulation of SIRT3, NCOA4, and the LC3-II/I ratio [57].

In addition to these mechanisms, puerarin also targets key pathways related to metastasis and cell adhesion. It downregulates the expression of C-C motif chemokine receptor 7 and C-X-C chemokine receptor type 4 (CCR7, CXCR4), MMP-2 and MMP-9, as well as intercellular adhesion molecule 1 (ICAM-1) and vascular cell adhesion molecule 1 (VCAM-1), ultimately inhibiting migration in MCF-7 and MDA-MB-231 cells. Furthermore, puerarin abrogates NF-κB activation in LPS-stimulated cells by suppressing phosphorylation of the NF-κB subunit proto-oncogene (p65) and inhibitor of nuclear factor kappa-B alpha (IκBα) [56].

## 4. Conclusions and Future Perspectives

The search for natural compounds constitutes a fundamental strategy for cancer treatment. Many chemotherapeutics currently in use are inspired by these compounds and act effectively on specific targets to inhibit tumor growth and metastasis. As previously described, isoflavones exhibit significant potential as antitumor agents.

Taken together, the data obtained from this ten-year literature survey allows a broader comparative interpretation of the available evidence. Across the different compounds, a consistent mechanistic pattern emerges: most isoflavones exert anticancer effects through apoptosis induction, ROS generation, cell cycle arrest, inhibition of migration and invasion, mitochondrial dysfunction, and in some cases antiangiogenic activity, with several agents also demonstrating tumor growth reduction in xenograft models. Breast and prostate cancer lines remain the most frequently used models, reflecting the relevance of estrogen-modulated pathways in isoflavone activity. Genistein and daidzein, the most extensively studied molecules, show the broadest mechanistic range, whereas more recently identified isoflavones present promising but still preliminary profiles. Although the heterogeneity of study designs and the predominance of single-study reports limit the feasibility of a deeper integrative mechanistic synthesis, the recurring pathways identified across compounds suggest shared molecular targets, while structural differences, particularly prenylation and glycosylation, appear to influence potency and cellular selectivity. These observations underscore the need for more comparative studies, especially for newly reported isoflavones, to determine how structural features correlate with anticancer effects and to advance the translational potential of this diverse class of molecules.

Furthermore, new molecular targets were identified through which isoflavones exert their effects, promoting cell death via the mechanisms mentioned above. Studies also revealed that isoflavones may act synergistically with each other or in combination with chemotherapeutic agents, enhancing antitumoral and cytoprotective activity. New drug delivery systems, such as nanoparticles and nanostructure-based complexes, improve therapeutic activity. Based on these findings, it can be inferred that both well-established isoflavones and those more recently described in the literature have demonstrated promising anticancer potential in both in vitro and in vivo studies.

However, further in-depth investigations of the newly identified isoflavones are still required. For most of these compounds, only a single study is available, typically limited to cytotoxicity assessments and analyses of cell death pathways. Thus, more comprehensive studies are necessary, including in vivo evaluations using xenograft models and detailed toxicological profiling. Such advances are essential to enable the transition of these compounds toward translational research applications.

## Figures and Tables

**Figure 1 biomedicines-13-02990-f001:**
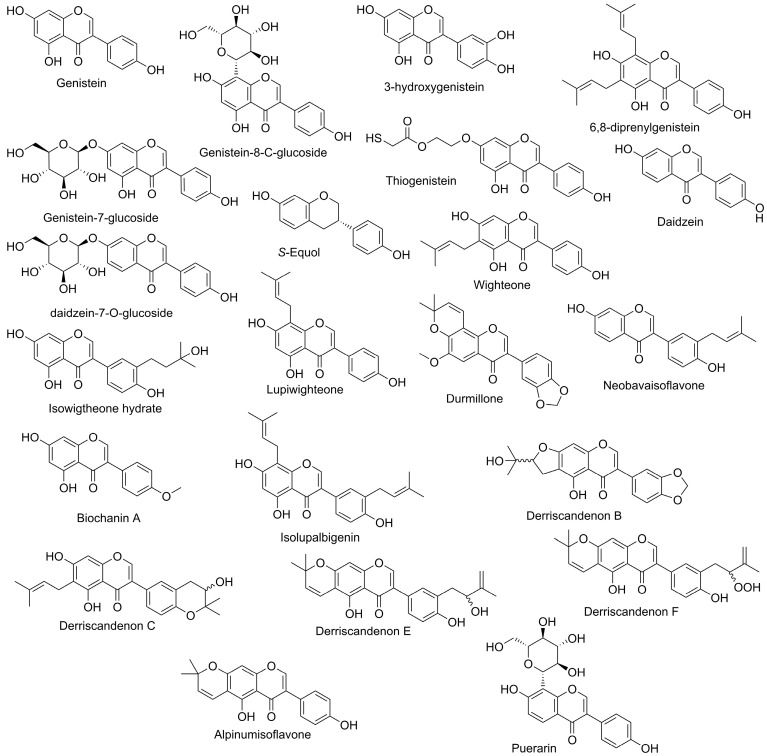
Schematic representation of the fundamental chemical scaffolds and substituent patterns that characterize the isoflavones examined throughout this review.

**Table 1 biomedicines-13-02990-t001:** Overview of antitumor activity reported for the isoflavones included in this review.

Antitumor Mechanism and/or Experimental Evidence	Animal/Cell Line Tested	IC50 (µM) ^a^, Selective Index (S.I.) ^b^; % Reference Growth Inhibition ^c^ or Dose ^d^	Ref.
Alpinumisoflavone			
Cytotoxicity (MTT), no mechanistic data available	PC-3 (prostate adenocarcinoma) SiHa (cervical carcinoma) A-549 (lung carcinoma) L929 (mouse fibroblasts, normal)	18.6 (PC-3); 19.6 (siHa); >303 (A-549); >297 (L929)-(IC50) ^a^. ~15.9 (PC-3); ~15.1 (SiHa) (S.I.) ^b^.	[14]
Cytotoxicity (MTT), no mechanistic data available	HL-60 (acute promyelocytic leukemia) SMMC-7721 (hepatocellular carcinoma) MCF-7 (breast adenocarcinoma) SW480 (colorectal adenocarcinoma)	0.98 (HL-60); 3.25 (SMMC-7721); 7.56 (MCF-7); 3.27 (SW480)-(IC50) ^a^.	[15]
Cytotoxicity (MTT), no mechanistic data available	K562 (leukemia) K562/ADR (leukemia)	42.7 (K562); 49.93 (K562/ADR)-(IC50) ^a^.	[16]
Apoptosis with changes in the mitochondrial membrane potential, with ROS generation; Cell migration inhibition; Upregulation of p53 and RRM1 proteins; Colony formation inhibition.	PANC-1 (pancreatic carcinoma) MIA PaCa-2 (pancreatic carcinoma)	28.92 (MIA PaCa-2); 35.82 (PANC-1)-(IC50) ^a^.	[17]
Biochanin A			
Cytotoxicity (MTT), with reduced EGFR activity	A549 (lung carcinoma) H1795 (lung carcinoma) HCC827 (lung carcinoma)	13.5 (A549); 42.0 (H1795); 0.135 (HCC827)-(IC50) ^a^.	[18]
Apoptosis via caspase-3 activation; Downregulation of ERα and Cyclin D1; Reduction in MMP-2 and VEGF proteins; Tumor growth inhibition (Ehrlich carcinoma)	MCF-7 (breast adenocarcinoma) MDA-MB-231 (breast adenocarcinoma) Balb/c mice (in vivo model)	1.97 (MCF-7); 4.87 (MDA-MB-231)-(IC50) ^a^. Tumor growth decrease at 20 mg/kg ^d^	[19]
Apoptosis; Decreased expression of phospho-AKT and phospho-mTOR; PARP cleavage; Reduced Cyclin A/B1 and STAT3 (Y705) activation	AsPC-1 (pancreatic adenocarcinoma) MIA PaCa-2 cells (pancreatic adenocarcinoma) PANC-1 (pancreatic carcinoma)	Biochanin A/atorvastatin (80 μM/5µM) ^d^. ~82%^c^ (AsPC-1 and MIA PaCa-2); ~35% ^c^ (PANC-1).	[20]
Cytotoxicity (MTT), no mechanistic data available; Induction of CYP1A1, CYP1B1, and UGT1A1 gene expression	Caco-2 (colorectal adenocarcinoma)	50 μM ^d^.	[21]
Daidzein			
Apoptosis with ROS generation; Cell migration inhibition; Upregulation of CTR1 and ATP7A genes	HPrEC (prostate epithelial cells, primary) LNCaP (prostate carcinoma) DU145 (prostate carcinoma)	50 µM ^d^. ~40%^c^ (LNCaP); ~45% ^c^ (DU145); ~10% ^c^ (HprEC). ~ 1.5 (LNCaP); ~1.6 DU145 (S.I.) ^b^.	[22]
Antigenotoxic and antiproliferative effects	LNCaP (human prostate cancer cell line) Sprague Dawley rats	10 µM ^d^. 70% of inhibition of 4NQO toxicity.	[23]
Decrease in caspase-3 activity; Colony formation inhibition	D283 Med (medulloblastoma)	10 nM ^d^ in combination with cisplatin	[24]
Apoptosis caspase-dependent; ROS generation; Upregulation of Bax; Downregulation of Bcl-2 and Erα and Increased Erβ expression	MCF-7 (breast adenocarcinoma)	100.0 (IC50) ^a^.	[25]
Cytotoxicity (MTT), with reduced EGFR activity	A549 (lung carcinoma) H1795 (lung carcinoma) HCC827 (lung carcinoma)	13.5 (A549); 42.0 (H1795); 0.135 (HCC827)-(IC50) ^a^.	[18]
Cytotoxicity (MTT), no mechanistic data available; Cell migration inhibition	A549 (lung carcinoma)	83.06 (IC50) ^a^.	[26]
Apoptosis with ROS generation; DNA damage pH-and Cu-dependent	MDA-MB-231 (breast adenocarcinoma)	50 (IC50) ^a^.	[27]
Cytotoxicity (Trypan blue), no mechanistic data available; Decreased β-catenin expression	HT-29 (colorectal adenocarcinoma)	100 (IC50) ^a^.	[28]
Apoptosis caspase-dependent, Cell migration and invasion inhibition; Decreased β-catenin, Cyclin D1, MMP9 and MPP2 expression; tumor growth inhibition	U2OS, HOS, MNNG/HOS, SJSA-1, 143B, MG63, U2OS/MTX300 (osteosarcoma cell line), male BALB/c nude mice	25.7 (U2OS); 58.7 (U2OS/MTX300); 56.9 (HOS); 33.4 (MNNG/HOS); 19.8 (SJSA-1) 26.4 µM (143B)-(IC50) ^a^. Tumor growth decrease with (20/5 mg/kg) of cisplatin and Daidzin ^d^.	[29]
Apoptosis with loss of mitochondrial membrane potential and with ROS generation; Cell cycle arrest at G0/G1 phase; Colony formation and migration inhibition	BGC-823 (endocervical adenocarcinoma) GES-1 (gastric epithelial cells)	19.64 (BGC-823); >60µM (GES-1)-(IC50) ^a^. ~3,1 (S.I.) ^b^.	[30]
Derriscandenon			
Apoptosis with loss of mitochondrial membrane potential	KB (epidermoid carcinoma) NALM-6 (acute lymphoblastic leukemia, B-cell) Fibroblasts (fibroblasts)	78.2%^c^ (NALM-6); 3.2% ^c^ (KB); > 25% ^c^ (fibroblasts). ~1.1. (NALM-6); ~7.8 (KB)-(S.I.) ^b^.	[31]
Apoptosis with loss of mitochondrial membrane potential	KB (epidermoid carcinoma) NALM-6 (acute lymphoblastic leukemia, B-cell) A549 (lung carcinoma) Fibroblasts (human dermal fibroblasts)	2.7 (KB); 9.5 (NALM-6); 23.4 (fibroblasts)-(IC50) ^a^. ~2.4 (NALM-6); ~8.6 (KB)-(S.I.) ^b^.	[32]
Durmillone			
Apoptosis with loss of mitochondrial membrane potential	MCF-7 (breast adenocarcinoma) HeLa (cervical adenocarcinoma) HUVECs (umbilical vein endothelial cells)	11.08 (MCF-7); 6.09 (HeLa); >50 (HUVEC)-(IC50) ^a^. ~4.5 (MCF-7); ~8.2 (HeLa)-(S.I.) ^b^.	[33]
Apoptosis with loss of mitochondrial membrane potential	U87MG (glioblastoma) MDA-MB-231 (breast adenocarcinoma) CEM/ADR5000 (acute lymphoblastic leukemia, T-cell)	0.86 (CEM/ADR500); 8.97 (MDA-MB-231); 5.83 (U87MG)-(IC50) ^a^.	[34]
Apoptosis with loss of mitochondrial membrane potential	A549 (lung carcinoma) BEAS-2B (bronchial epithelial cells) LO2 (fetal hepatocytes) CCD19Lu (fibroblasts)	6.6 (A549); 58.4 (BEAS-2B); 78.4 (LO2); >100 (CCD19Lu)-(IC50) ^a^. ~15.1 (S.I.) ^b^.	[35]
Equol			
Antigenotoxic and antiproliferative effects	LNCaP (prostate carcinoma) Sprague Dawley rats (in vivo model)	10 µM ^d^ + 0.25 mg/L of 4NQO and 1 mg/L to 5 mg/L ^d^ of 2AA with 55 ± 9% genotoxicity inhibition ^c^.	[23]
Apoptosis with FasL and Bim protein expression; Cell cycle arrest at G2/M phase; CDK1 and Cyclin B inhibition; Increased FOXO3a, p21, p27 and decreased p-FOXO3a expression; Antitumoral activity in xenografts of PC3 cells	PC-3 (prostate adenocarcinoma) DU145 (prostate carcinoma) LNCaP (prostate carcinoma) RWPE-1 (prostate epithelial cells, non-malignant) BALB/c mice (in vivo model)	119 (PC-3); 139 (DU145); 169 µM (LNCaP)-(IC50) ^a^. Inhibited tumor growth 43.2% ^c^ at 20 mg/Kg ^d^	[36]
Decrease in caspase 3 activity and cell colony formation inhibition	D283 Med MB (medulloblastoma)	100 nM with 10 nM of cisplatin ^d^	[24]
Apoptosis with downregulation of Bcl-xL expression and activation of Akt and mTOR; Cell Cycle Arrest at G2/M	MCF-7 (breast adenocarcinoma) MDA-MB-468 (breast adenocarcinoma) SK-BR-3 (breast adenocarcinoma)	56.8 (MCF-7); 67.0 MDA-MB-468); 42.1 (SK-BR-3)-(IC50) ^a^.	[37]
Genistein			
Apoptosis and negative regulation of Erα	ER-positive MCF-7 (breast adenocarcinoma) ER-negative MDA-MB-231 (breast adenocarcinoma)	68.34% ^c^ (MCF-7); 47.37% ^c^ (MDA-MB-231)	[38]
Cytotoxicity (MTT) no mechanistic data available	KB cell (carcinoma epidermoid) and HepG2 (hepatocellular carcinoma)	26.99 (KB) 19.5 (HepG2)-(IC50) ^a^.	[39]
Apoptosis caspase-dependent; VEGFA, MMP-9 downregulation; Antitumoral effect in chemically induced Hepatocellular carcinoma	HepG2 (hepatocellular carcinoma) Swiss albino mice (in vivo model)	23.6 (IC50) ^a^. 75 mg/kg to 100 mg/kg ^d^.	[40]
Apoptosis with loss of mitochondrial membrane potential and ROS generation	SK-OV-3 (ovarian carcinoma)	90.0 (IC50) ^a^.	[41]
Cytotoxicity (MTT), with reduced EGFR activity	A549 (lung carcinoma) H1795 (lung carcinoma) HCC827 (lung carcinoma)	13.5 (A549); 42.0 (H1795); 0.135 (HCC827)-(IC50 ^a^).	[18]
Apoptosis with p53 activation, ROS generation, and with TOPII-inhibiting potential	HT-29 (colorectal adenocarcinoma)	200 (IC50) ^a^.	[42]
Apoptosis caspase-depedent, Cell viability, Downregulation Erα of BcL2/BcLx-xL, MPP9 and cyclin D1	MCF-7 and MDA-MB-231 cells (Human breast carcinoma)	150 µM ^d^ with 60% ^c^ (MCF-7); 35% ^c^ (MDA-MB-231 cells).	[43]
Apoptosis with ROS-generation; Inhibition of cancer cell migration; Upregulation of CTR1 and ATP7A expression	HPrEC (prostate epithelial cells, primary) LNCaP (prostate carcinoma) DU145 (prostate carcinoma)	50 µM^d^ with ~40% ^c^ (LNCaP); ~45% ^c^ (DU145); ~10% ^c^ (HprEC). ~ 1.5 (LNCaP); ~1.6 DU145 (S.I.) ^b^.	[22]
Antigenotoxic and antiproliferative effects	LNCaP (prostate carcinoma) Sprague Dawley rats (in vivo model)	10 µM ^d^ + 0.25 mg/L of 4NQO and 1 mg/L to 5 mg/L ^d^ of 2AA with 55 ± 9% ^c^ genotoxicity inhibition.	[23]
Decrease in caspase 3 activity; Colony formation inhibition	D283 Med MB (medulloblastoma)	10 and 100 nM ^d^	[24]
Cytotoxicity (WST), no mechanistic data available; Cell Cycle Arrest at G2/M	HAG/src31 (adenocarcinoma, v-src-transfected) HAG/neo3-5 (epithelial cell line, HAG-1 derivative)	25.0 (HAG/src31); 50.0 HAG/neo3-5-(IC50) ^a^.	[44]
Apoptosis non-mitochondrial pathway; Reduction in NO; Increase expression of catalase and glutathione (GSH)	PC3 (prostate adenocarcinoma)	~200 (IC50) ^a^.	[45]
Cytotoxicity (PrestoBlue assay), no mechanistic data available	DU145 (prostate carcinoma)	100.0 (IC50) ^a^.	[46]
Apoptosis with downregulation of BAX and BCL-xl genes; Cell cycle Arrest at G1 phase	MCF-7 (breast adenocarcinoma) HB4a (mammary epithelial cells)	100.0 (IC50) ^a^. Cell Cycle arrest 25µM ^d^	[47]
Cytotoxicity (Trypan blue), no mechanistic data available; Decrease in β-catenin gene (CTNNBIP1) expression	HT-29 (colorectal adenocarcinoma)	100 (IC50) ^a^.	[28]
Cytotoxicity (MTT), no mechanistic data available; Cell Cycle Arrest at G1/S phase; downregulation of mTOR and of cyclin E1; Antitumoral activity in xenografts of Y70 cells	Y79 (retinoblastoma) BALB/c nude mice (in vivo model)	1.23 µM (IC50) ^a^. 73.24% tumor weight reduction ^c^ with 60 mg/kg ^d^	[48]
Cytotoxicity (MTT), no mechanistic data available; Antiangiogenic effects	HeLa (cervical adenocarcinoma) A431 (epidermoid carcinoma) MCF-7 (breast adenocarcinoma) A2780 (ovarian carcinoma)	69.88 (A431); 44.02 (MCF-7); 17.49 (A2780); 16.47 (HeLa)-(IC50) ^a^. antiangiogenic effect with 50 µM ^d^	[49]
Apoptosis with downregulation of Bcl-xL expression and upregulation of Akt and mTOR; Cell Cycle Arrest at G2/M	MCF-7 (breast adenocarcinoma) MDA-MB-468 (breast adenocarcinoma) SK-BR-3 (breast adenocarcinoma)	56.8 (MCF-7); 67.0 MDA-MB-468); 42.1 (SK-BR-3) (IC50) ^a^.	[37]
Cytotoxicity (MTT), no mechanistic data available; Decreases cell migration; Downregulation of miR-155 and HK2 expression	MCF7 TAM-S (breast adenocarcinoma, tamoxifen-sensitive) MCF7 TAM-R (breast adenocarcinoma, tamoxifen-resistant)	20.0 (MCF-7 TAM-S); >20.0 µM MCF7 (TAM-R) (IC50) ^a^.	[50]
Apoptosis with ROS-generation; Cell migration inhibition; Cell cycle arrest at S and G2 phase. Antitumoral activity in xenografts of Bel-7402 cells	LO2 (liver cells) HUVECs (umbilical vein endothelial cells) Bel-7402 (endocervical adenocarcinoma) BALB/c nude mice (in vivo model)	1.42 µM (Bel-7402); >80 (HUVECs and LO2) (IC50) ^a^. 20 mg/kg ^d^ with 3.7-fold tumor reduction	[51]
Isolupalbigenin			
Cytotoxicity (MTT), no mechanistic data available	K562 (leukemia) K562/ADR (leukemia) A549 (lung carcinoma) SPC-A-1 (lung carcinoma) ASG (gastric carcinoma) HeLa (cervical adenocarcinoma) HepG2 (hepatocellular carcinoma) CNE2 (nasopharyngeal carcinoma) HaCaT (normal epidermal keratinocytes)	56.44 (AGS); 1.55 (K562); 50.2 3(K562/ADR); 52.40(HeLa); 52.67; (HepG2), 52.86; (CNE2); 49.86; (SPC-A-1) 33.41 (Hacat)-(IC50) ^a^. 21.5 (K562)-(S.I.) ^b^.	[16]
Apoptosis caspase-dependent and inhibition of Glyoxalase I (GLO I)	HL-60 (leukemia)	13.4 (IC50) ^a^.	[52]
Apoptosis and colony formation inhibition	HepG2 (hepatocellular carcinoma)	40 (IC50) ^a^.	[53]
Neobavaisoflavone			
Apoptosis with changes in the mitochondrial membrane potential; Cell cycle arrest at G1/S phase	SW1783 (anaplastic astrocytoma)	75.0 (IC50) ^a^.	[54]
Apoptosis with changes in the mitochondrial membrane potential; Cell cycle arrest at G1 phase	U-87 MG (glioblastoma) NHA (normal human astrocytes)	50.0 (IC50) ^a^.	[55]
Puerarin			
Cytotoxicity (Cell Counting Kit-8), no mechanistic data available; NF-kB inhibition; with downregulation of CCR7, CXCR4, MMP-2, MMP-9, ICAM and VCAM; Inhibition cell migration	MCF-7 (breast adenocarcinoma) MDA-MB-231 (breast adenocarcinoma)	80.0 (IC50) ^a^.	[56]
Cytotoxicity (Cell Counting Kit-8) with ROS generation; Decreased of GSH/GSSG ratio; Upregulation of SIRT3, NCOA4, and the ratio of LC3-II/I	U87MG (glioblastoma)	40.0 (IC50) ^a^.	[57]
Apoptosis caspase-dependent	HeLa (cervical adenocarcinoma)	25.0 (IC50) ^a^.	[58]
Wighteone			
Cytotoxicity (MTT), no mechanistic data available	K562 (leukemia) ASG (gastric carcinoma)	55.46 (ASG); 50.28 (K562)-(IC50) ^a^.	[16]
Apoptosis caspase-dependent with ROS generation	HepG2 (hepatocellular carcinoma) HL-60 (promyelocytic leukemia) A549 (lung carcinoma) HeLa (cervical adenocarcinoma) KB (epidermoid carcinoma) HT29 (colorectal adenocarcinoma) LO2 (normal hepatic cells)	3.2 (HepG2); 75.1 (LO2); 14.9 (HeLa); 51.4 (HT-29); 73.3 (A-549); 14.3 (KB);2.5 (HL-60) (IC50) ^a^. 23.5 (HepG2) (SI) ^b^.	[59]
Cytotoxicity (MTT), no mechanistic data available	PC-3 (prostate adenocarcinoma)	34.3 (IC50) ^a^.	[14]
Apoptosis with changes in the mitochondrial membrane potential and ROS-generation; Inhibition of CDK1 and CDK2; Downregulation of the Akt and VEGF; Cell cycle arrest at G2/M phase; Cell migration inhibition.	DU-145 (prostate carcinoma) HUVECs (umbilical vein endothelial cells)	23.7(DU-145); 50.5 (HUVEC)-(IC50) ^a^. 2.1 (S.I.) ^b^.	[60]

The table is subdivided by compound. Each row corresponds to a single reference. Columns are organized as follows: (i) Antitumor mechanism and/or experimental evidence, describing the specific cellular or molecular effects reported; (ii) Cell lines tested and, when applicable, animal models; and (iii) Main findings reported in the original publication. ^a^ IC_50_: concentration required to inhibit 50% of cell viability or proliferation. ^b^ Selectivity index (SI): ratio between IC_50_ in tumor cells and IC_50_ in normal cells (reported as exact or approximate values). ^c^ Percentage of inhibition: reduction in cell viability or proliferation at a specified concentration. ^d^ Dose: in vitro, refers to the concentration that produced the highest growth-inhibitory effect; in vivo, refers to the amount of compound administered per kilogram that resulted in measurable antitumor activity.

## Data Availability

The data used in this article were sourced from the materials mentioned in the References section.

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
