# Peer review of "Anticancer Potential of Isoflavones: A Narrative Overview of Mechanistic Insights and Experimental Evidence from the Past Ten Years"

_biomedicines, 2025, doi:10.3390/biomedicines13122990_

Round 1
Reviewer 1 Report
Comments and Suggestions for Authors
A major revision of the manuscript is necessary before the manuscript is recommended for publication in Biomedicines.

The English could be improved to more clearly express the research.
Author Response
“After a thorough review of the manuscript titled "Anticancer Potential of Isoflavones: Mechanistic Insights and Experimental Evidence from the Past Ten Years," I commend the authors for undertaking a comprehensive and timely review of a highly relevant topic in natural product drug discovery. The manuscript is well-structured, covers a vast body of literature, and provides a valuable summary of the anticancer mechanisms of various isoflavones. The inclusion of a detailed table (Table 1) and a chemical structures figure enhances the clarity and utility of the review. However, to elevate the manuscript from a descriptive summary to a more critical and impactful scholarly work, several aspects require significant refinement, particularly regarding the analytical depth, methodological transparency, and overall narrative. The following points should be addressed in a major revision.”
Reply: Thank you for considering the impact and relevance of our work.
1) Critical Analysis and Synthesis Beyond Description
The review excels at cataloging individual studies but falls short of providing a critical synthesis of the evidence. The narrative is largely descriptive, listing mechanisms (e.g., "induces apoptosis," "generates ROS") for each isoflavone without a comparative analysis. The authors should integrate the findings to address overarching questions: What are the most consistently reported and potent mechanisms across different isoflavones and cancer types? Are there apparent structure-activity relationships (e.g., how prenylation or glycosylation affects potency or mechanism)? A deeper, comparative discussion that identifies patterns, contrasts divergent findings, and highlights the most promising mechanistic avenues would significantly strengthen the scholarly contribution.
Reply: Although we fully acknowledge the relevance of the reviewer’s suggestion regarding a deeper comparative and mechanistic synthesis, the current state of the available evidence imposes important limitations. Most of the isoflavones included in this review have been investigated in only one or very few studies, frequently restricted to in vitro cytotoxicity assays using isolated cancer cell lines. As a result, the available data do not provide sufficient consistency, reproducibility, mechanistic depth, or structural diversity to support a robust cross-comparison or meaningful structure–activity relationship analysis without risking overinterpretation.
For this reason, a broader integrative synthesis, such as identifying the “most potent mechanisms,” comparing effects across cancer types, or drawing conclusions on the role of prenylation or glycosylation, would not be methodologically sound at this stage, given the heterogeneity and scarcity of the evidence. We therefore opted to preserve the descriptive approach in the main body of the review, as it more accurately reflects the fragmented and preliminary nature of current findings.
However, fully aligned with the reviewer’s concern, we strengthened the “Conclusions and Future Perspectives” section to explicitly address these limitations (as highlighted and below), highlight the unmet need for systematic investigations, and outline the research gaps that must be filled before a higher-level mechanistic synthesis becomes feasible. We believe this adjustment improves the manuscript while avoiding speculative conclusions that are not adequately supported by the existing literature.
New part of the conclusion: “However, further in-depth investigations of the newly identified isoflavones are still required. For most of these compounds, only a single study is available, typically limited to cytotoxicity assessments and analyses of cell death pathways. Thus, more comprehensive studies are necessary, including in vivo evaluations using xenograft models and detailed toxicological profiling. Such advances are essential to enable the transition of these compounds toward translational research applications.”
- Methodological Rigor and Reproducibility of the Literature Search
The methodology section is overly simplistic and lacks the detail required for reproducibility, which is a cornerstone of systematic reviewing. The authors must specify whether this is a systematic review or a narrative review. The search strategy should be detailed using the PRISMA guidelines or similar, including the exact search query with Boolean operators, the number of records identified, screened, and included, and the criteria for study selection (e.g., exclusion of non-English papers, types of studies). A flow diagram is highly recommended. Furthermore, the rationale for focusing only on studies with "more than two independent references" is not sufficiently justified and may introduce a selection bias, potentially overlooking novel, high-impact findings from single studies.
Reply: We thank the reviewer for this important observation. We have substantially revised the writing of our Materials and Methods section. But we would like to clarify that the present study is a narrative literature review, not a systematic review. Therefore, the methodology applied follows the standards commonly adopted for narrative reviews, in which the aim is to present a broad, integrative, and descriptive synthesis of the available evidence rather than to perform a reproducible systematic screening following PRISMA guidelines. To avoid misunderstanding, we explicitly added the term “Narrative Overview” to the title of the manuscript:
“Anticancer Potential of Isoflavones: A Narrative Overview of Mechanistic Insights and Experimental Evidence from the Past Ten Years.”
The Methodology section was expanded to clearly describe the type of review, search strategy, and criteria used, following the structure of recent narrative reviews published in the same journal (DOI: 10.3390/biomedicines13112802; 10.3390/biomedicines13112731).
Regarding the inclusion criteria, we acknowledge the reviewer’s concern about the focus on studies with “more than two independent references.” Our rationale was based on the large volume of isolated reports and the low representativity of findings described only once in the literature, which, if treated equally in the main text, would hinder thematic cohesion and obscure mechanistic patterns. To address this limitation transparently, all studies supported by only a single reference were nevertheless catalogued, classified, and listed, along with their corresponding antitumor activities and citations, in the Supplementary Material.
This approach ensures that:
No available evidence was excluded from the manuscript, maintaining completeness;
The main text remains focused on the most consistently studied isoflavones, improving interpretative clarity;
Readers still have full access to all single-report findings in a dedicated supplementary section, ensuring transparency and avoiding potential selection bias.
We have substantially revised the writing of our Materials and Methods section as below:
“2.1. Type of Review
This work is a narrative literature review, designed to synthesize recent experimental evidence on the anticancer properties of isoflavones. Given the wide heterogeneity of study designs, biological models, and mechanistic endpoints reported in the literature, a narrative review framework was considered the most appropriate approach for summarizing current knowledge without imposing the methodological constraints of a systematic review.
2.2. Search Strategy
The literature survey was conducted using the PubMed database and covered the period January 2014 to April 2025. The search strategy used the keywords “isoflavones” combined with at least one additional term (“anticancer,” “antitumoral,” “antiproliferative,” or “cytotoxicity”) appearing in the title or abstract. This strategy allowed us to identify studies specifically investigating experimentally validated anticancer activity.
2.3. Study Selection Process
Studies were included when they provided experimental in vitro and/or in vivo evidence of the antitumor activity of structurally elucidated isoflavones. Articles were excluded when they:
(1) described only in silico results;
(2) did not directly assess anticancer activity;
(3) examined compounds whose chemical structures had not been elucidated; or
(4) were written in languages other than English.
Because many newly identified isoflavones were reported in single, isolated publications, including highly heterogeneous methodologies, compiling them directly into the main dataset would reduce comparability and distort the identification of recurrent mechanisms. Nonetheless, to ensure transparency and avoid selection bias, all such single-reference isoflavones were catalogued and listed in the Supplementary Material, including their reported biological activities and corresponding references.
2.4. Data Analysis
For the narrative synthesis presented in the main text, we conducted a more detailed descriptive analysis only for isoflavones that had been investigated in two or more independent publications. This criterion allowed us to extract mechanisms that were recurrent or at least independently reproduced, avoiding overinterpretation based on isolated findings.
For these compounds, we summarized experimental data including:
– antitumor mechanisms;
– gene and protein expression modulation;
– effects on cell death pathways;
– cell line models used;
– in vivo xenograft results (when available);
– and overall biological outcomes in relation to the methodological approaches.
Language editing was supported using the TRINKA grammar tool; all text generated or modified through this tool was subsequently reviewed, edited, and finalized by the authors, who take full responsibility for the content.”
- Assessment of Evidence Quality and Translational Potential
The review presents a wealth of in vitro data but does not critically appraise the quality of the evidence or the clinical relevance of the effective concentrations. Many reported IC50 values are in the high micromolar range (e.g., 50-100 µM for daidzein), which may not be pharmacologically achievable in humans. The authors should discuss the pharmacokinetic and bioavailability challenges of isoflavones and contextualize the in vitro findings within these limitations. Additionally, for the in vivo studies mentioned, a critical evaluation of the model robustness (e.g., xenograft vs. syngeneic models), dosing regimens, and how they translate to potential human doses is necessary to assess the true translational potential of these compounds.
Reply: We appreciate the suggestion. We believe that a translational analysis could be the focus of a separate narrative review. In our case, the scope defined by the authors is limited to describing the mechanisms and the experimental evidence of the antitumor activity of isoflavones over the past ten years. The vast majority of the published studies concern newly isolated substances from plant extracts, for which only preliminary in vitro assays have been conducted. A translational analysis would be feasible only if more robust data were available. Within the literature surveyed, we found few studies that performed preclinical assays in animals, which would be the type of evidence most closely aligned with a translational approach. To take this step, variables such as the toxicity of the described substances, lethal dose, pharmacokinetic and pharmacodynamic evaluation of the substance among others, would need to be considered. Unfortunately, these data remain scarce.
- Analysis of Conflicting Evidence and Dual Effects of Isoflavones
The manuscript occasionally mentions but does not sufficiently explore contradictory findings, which is a critical aspect of a balanced review. A prime example is the dual role of genistein and equol, which can act as cytoprotective agents in medulloblastoma cells by inhibiting cisplatin-induced cytotoxicity. This contrasts sharply with their pro-apoptotic roles in other cancers. The review needs a dedicated section discussing these context-dependent effects, exploring potential reasons such as cancer type-specific signaling pathways, hormonal receptor status, and concentration-dependent effects. Failing to adequately address these contradictions presents an incomplete picture of isoflavone pharmacology.
Reply: In this article, our goal was to describe the antitumor activity of each isoflavone. Thus, each compound is presented as a separate subtopic. In order to incorporate the reviewer’s suggestion without disrupting the structural organization of the text, We added paragraphs highlighting the relevance of the dual effect, with emphasis on the study that reports the cytoprotective activity (inside genistein subtopic):
“Dual effect of isoflavones when combined with clinically used chemotherapeutic agents is particularly relevant. Their ability to interact with estrogen receptors represents a mechanism capable of enhancing antitumor activity, since this interaction may inhibit proliferative signaling pathways that are not targeted by the accompanying compound. Moreover, the combined action of multiple isoflavones may further potentiate this interaction, as one isoflavone may exhibit greater affinity for certain estrogen receptor subtypes than another, thereby increasing the likelihood of an effective mechanism of action. It has also been demonstrated that, particularly those soy-derived, isoflavones exhibit chemopreventive properties, such as anti-inflammatory and antioxidant activities in breast cancer models, while simultaneously modulating the expression of genes involved in cell-cycle progression and apoptosis activation (as detailed throughout the text) in the same model [10, 29, 30]. Together, these findings reinforce the relevance of this dual activity.
On the other hand, estrogens may exert cytoprotective effects, as observed in medulloblastoma, where they can reduce sensitivity to the cytotoxic effects of cisplatin, given that tumor cell growth is stimulated by 17β-estradiol. This phenomenon may increase chemoresistance. Therefore, inhibition of the estrogen receptor ERβ may enhance treatment efficacy and suppress the tumor growth induced by exogenous estradiol. Considering the anti-estrogenic properties of isoflavones, they may inhibit this target, acting as adjuvants to chemotherapy. Genistein exhibits a cytoprotective effect against cisplatin-induced cytotoxicity and inhibits colony formation in the medullo-blastoma cell line D283 Med.”
Regarding the data on hormonal interactions, pharmacological aspects, concentrations, and related parameters, these details are not available in all the studies included in our selection. Therefore, this would not result in a fully comprehensive or consistent subsection.
5. Structural Organization and Thematic Flow
The current organization by individual isoflavone (e.g., 3.1. Genistein, 3.2. Daidzein) leads to significant repetition, as similar mechanisms (apoptosis, ROS, cell cycle arrest) are described separately for each compound. This structure impedes the development of a cohesive narrative. It is recommended to reorganize the "Anticancer Activity" section into thematic subsections based on primary mechanisms of action (e.g., "Apoptosis Induction," "Cell Cycle Arrest," "Anti-metastatic Effects," "Modulation of Hormone Signaling"). Within each subsection, the effects of different isoflavones should be compared and contrasted. This thematic approach would facilitate a more analytical and less repetitive discussion.
Reply: We sincerely appreciate this suggestion and fully agree that a thematic mechanistic reorganization is commonly appropriate in broader systematic or scoping reviews. However, after careful consideration, we believe that restructuring the manuscript in this manner would not be feasible without fundamentally altering the nature, scope, and objective of the present work.
This review was designed to provide a compound-centered overview, allowing readers to identify, for each individual isoflavone, the specific experimental evidence, test systems, and mechanistic findings available to date. Because the majority of isoflavones have only one or few published studys reporting anticancer effects, a thematic restructuring would lead to fragmented subsections populated by isolated, non-comparable findings and, consequently, to a high risk of overgeneralization. In several mechanisms (e.g., autophagy, angiogenesis, hormone signaling), only one compound has been tested, making comparative synthesis impossible without extrapolation.
To minimize redundancy while maintaining scientific accuracy, we revised the “Conclusions and Future Perspectives” (below) section to explicitly address the recurrent mechanisms observed across compounds and to discuss overarching patterns without reorganizing the entire Results section. We believe this solution strengthens the analytical component of the manuscript while preserving the structure that most faithfully reflects the current fragmentary state of the evidence.
We hope the reviewer finds this justification satisfactory.
New added conclusion:
“Taken together, the data obtained from this ten-year literature survey allows a broader comparative interpretation of the available evidence. Across the different com-pounds, a consistent mechanistic pattern emerges: most isoflavones exert anticancer ef-fects through apoptosis induction, ROS generation, cell-cycle arrest, inhibition of migra-tion and invasion, mitochondrial dysfunction, and in some cases antiangiogenic activity, with several agents also demonstrating tumor growth reduction in xenograft models. Breast and prostate cancer lines remain the most frequently used models, reflecting the relevance of estrogen-modulated pathways in isoflavone activity. Genistein and daidzein, the most extensively studied molecules, show the broadest mechanistic range, whereas more recently identified isoflavones present promising but still preliminary profiles. Although the heterogeneity of study designs and the predominance of single-study re-ports limit the feasibility of a deeper integrative mechanistic synthesis, the recurring pathways identified across compounds suggest shared molecular targets, while structural differences, particularly prenylation and glycosylation, appear to influence potency and cellular selectivity. These observations underscore the need for more comparative studies, especially for newly reported isoflavones, to determine how structural features correlate with anticancer effects and to advance the translational potential of this diverse class of molecules.”
6. Contextualization within the Current Anticancer Drug Discovery Landscape
The introduction and conclusion would benefit from a broader perspective on the role of natural products like isoflavones in modern oncology. The authors should discuss how the multifaceted mechanisms of isoflavones (e.g., targeting EGFR, modulating copper homeostasis) align with current targeted therapy trends. Furthermore, the conclusion should go beyond summarizing findings to propose specific, actionable future research directions. These could include the need for more sophisticated drug delivery systems to overcome bioavailability issues, the exploration of isoflavones as chemosensitizers in combination therapies, and the importance of clinical trials to validate preclinical promises.
Reply: In both the Introduction and the Conclusions and Future Perspectives section, we incorporated an opening paragraph before the final conclusion to highlight the relevance of natural products. In the Introduction, we added the following contextualization to reinforce the historical and pharmacological importance of natural compounds in anticancer drug discovery:
“Natural products have attracted considerable interest from the pharmaceutical industry, particularly due to the advances achieved over the past 50 years. Notably, antitumor therapy has made significant progress as a result of these discoveries[3] . Consequently, a large proportion of antineoplastic therapies approved in recent decades have a direct or indirect origin in natural products, whether through isolated molecules, semisynthetic derivatives, or compounds inspired by their chemical structures[4]. The most well-known examples are the vinca alkaloids (vincristine and vinblastine) and Paclitaxel, which was isolated from the plant species Taxus brevifolia [5]. These compounds frequently act on key cellular signaling pathways, including microtubule formation, DNA synthesis, apoptosis, oxidative stress, the inhibition of cell-cycle progression, angiogenesis and cell migration [6]. Isoflavones are a remarkable example of a class of natural products with antitumor potential.”
Similarly, in the Conclusions and Future Perspectives section, an introductory statement was inserted to underscore how isoflavones fit within the broader context of natural‐product–based anticancer research:
“The search for natural compounds constitutes a fundamental strategy for cancer treatment. Many chemotherapeutics currently in use are inspired by these compounds and act effectively on specific targets to inhibit tumor growth and metastasis. As previously described, isoflavones exhibit significant potential as antitumor agents.”
Additionally, to ensure that the closing remarks reflect current therapeutic trends, we incorporated the following statement highlighting combined strategies and innovative delivery systems:
“…in combination with chemotherapeutic agents, enhancing antitumoral and cytoprotective activity. New drug delivery systems, such as nanoparticles and nanostructure-based complexes, improve therapeutic activity.”
7. Data Presentation and Consistency in Tables and Figures
While Table 1 is a valuable resource, its formatting and consistency need improvement. The use of complex, nested cell merging makes it difficult to read and follow. The table should be simplified for clarity, with each row representing a unique compound-study pairing. Abbreviations should be defined in a dedicated footnote, not within the table cells, to improve readability. The chemical structures figure (Figure 1) is essential, but its quality and integration into the text should be verified during production. The authors should ensure all structures are clearly drawn and labeled, and the figure caption should explicitly state that it shows the core structures of the isoflavones discussed in the review
Reply: We appreciate the reviewer’s comments regarding Table 1 and Figure 1. Table 1 follows the editorial standards previously accepted by the journal and will undergo professional formatting during the production stage, which typically enhances clarity, alignment, and visual consistency. However, we made a full revision of the Table and now we think that it´s clearer and more simplified.
Each row in the table corresponds to a unique bibliographic reference, rather than an individual compound–activity pairing. This structure is necessary because a single isoflavone is often evaluated across multiple biological activities within the same study. If each compound–activity combination was separated into independent rows, the table would become disproportionately long and fragmented, reducing rather than improving readability. The current format was intentionally designed to maintain transparency and facilitate direct tracing of each activity back to its original publication.
All chemical structures displayed in Figure 1 are correctly drawn, labeled, and nomenclated. A high-resolution version of this figure will be provided in the final submission to ensure optimal graphical quality during typesetting. Additionally, all abbreviations used throughout the table are already defined in the dedicated Abbreviations section of the manuscript; transferring them to footnotes would duplicate information without adding clarity. Finally, removing tumor-origin details—as suggested—would reduce the contextual value of the table and impair the reader’s ability to interpret biological relevance across distinct cancer models.
Finally, figure caption was rewritten as follow: “Representation of the core structures of the isoflavones addressed throughout this review”
Reviewer 2 Report
Comments and Suggestions for Authors
Overview:
Isoflavones are phytoestrogens commonly found in soybeans. In this review Knupp et al. summarize the anticancer potential of isoflavones reported over the past decade, including both vitro and in-vivo data, with a special emphasis on the reported anticancer mechanisms. The review provides valuable information; however, it requires further improvement before it can be considered for publication.
- Please proofread the manuscript carefully to correct grammar and punctuation errors. Several sentences start or end abruptly, making them difficult to follow. Additionally, verb tenses need improvement for consistency and clarity.
- Abstract: Line 26 – The statement “Reported mechanisms include ….. inhibition of migration, antitumor xenograft studies” is scientifically inaccurate. Antitumor xenograft studies represent an experimental approach, not a mechanism of action.
- Page 2, Line 68 – The statement “Particularly regarding its anticancer potential, as previously demonstrated in earlier studies” is not a complete sentence, please revise.
- Table 1 requires significant reorganization to improve clarity and readability. Please reconsider the structure so that the information is easier for readers to interpret. Currently, the term “cytotoxicity” is used broadly; however, it simply refers to the property of being toxic to cells. To enhance the scientific value of the table, please revise the table to focus on the specific mechanisms by which these compounds exert cytotoxic effects. Additionally, verify all entries against the original sources, as several inconsistencies were noted. The following points represent a non-exhaustive list of issues observed:
- Cytotoxicity is incorrectly spelled as ‘Citotoxicity’.
- Reference 23 – Please verify the IC₅₀ value listed (17 µM) against the original publication. The reported values in the source do not appear to match. Ensure accuracy and consistency with the cited reference.
- Reference 35- The statement “The expression of the CYP24A1 gene induction is strongly amplified by hypoxia, the typical microenvironment of solid tumors” does not describe the anti-cancer mechanism of Genistein. Please revise.
- Reference 25 – “antitumor xenograft studies”. Antitumor xenograft studies represent an experimental approach, not a mechanism of action.
- Reference 53 – “Tumor growth inhibition (Ehrlich carcinoma)”. Not an anticancer mechanism. Please revise.
- Please add a separate section highlighting compounds that have been evaluated in in-vivo studies. This will help readers distinguish between preclinical evidence and cell-based data.
- Page 9, Lines 117 and 122. The sentences “One possibility is that genistein may suppress Src-driven proliferative activity” and “Some derivatives also have great anticancer activity” are not scientifically rigorous. Please revise to provide clear, evidence-based statements and specify mechanisms or pathways supported by data.
- Page 9, Line 134 to 37 and Page 10, Line 183 – The sentences describing cyclodextrin and phytosome formulations of genistein would fit better in the paragraph at Line 138, which focuses on different formulations/delivery systems. In general, please restructure this section to consolidate all delivery systems into a single paragraph for improved flow and coherence. This will also eliminate redundant discussion of the same delivery system across multiple paragraphs in the current version.
- Page 14, Lines 352 and 354 – Please add a sentence explaining the difference between Wighteone, Lupiwighteone, and Isowighteone.
- The authors should consider adding a table that describes and distinguishes the natural and synthetic isoflavones discussed in the manuscript.
- Page 10, Line 155: The sentence “Genistein downregulates the expression of estrogen receptor… (Bax/BCL-2)” is unclear and scientifically incorrect. Please revise for clarity and accuracy.
- Page 10, Line 171: The sentences “Using 3D culture models, it triggers apoptosis in a dose-dependent manner in PC-3 cells [13]’’ and “The downregulation of microRNA-155 (miR-155) and its target genes related to apoptosis and glucose metabolisms, such as STAT3 and hexokinase 2, particularly in tamoxifen-sensitive 174 MCF-7 cells [12]” are unclear and scientifically incorrect. Please revise for clarity and accuracy.
- Page 13, Line 314: The sentence “The tumor volume was drastically reduced by ~90% with the combination of daidzein and carboplatin” is inaccurate. The reference paper uses cisplatin, and not carboplatin. These are two different compounds.
- Page 15, Line 392: Typo ‘coumpond’ instead of ‘compound’.
Author Response
“Isoflavones are phytoestrogens commonly found in soybeans. In this review Knupp et al. summarize the anticancer potential of isoflavones reported over the past decade, including both vitro and in-vivo data, with a special emphasis on the reported anticancer mechanisms. The review provides valuable information; however, it requires further improvement before it can be considered for publication.”
Thank you for considering the relevance of our work. We have made a considerable effort to explore the topic in depth
1. Please proofread the manuscript carefully to correct grammar and punctuation errors. Several sentences start or end abruptly, making them difficult to follow. Additionally, verb tenses need improvement for consistency and clarity.
Reply: Since we had only one week to complete the revision, we were unable to hire a professional whose fees fit within our budget to perform the English-language proofreading. However, aiming to improve the grammatical quality of the manuscript, we used the TRINKA Grammar Checker, an online tool, to correct errors that could potentially compromise the meaning of the text. It´s use was made clear in the Material em Methods Section.
2. Abstract: Line 26 – The statement “Reported mechanisms include ….. inhibition of migration, antitumor xenograft studies” is scientifically inaccurate. Antitumor xenograft studies represent an experimental approach, not a mechanism of action.
Reply: I rewrote it in a way that separates the mechanisms from the experimental evidence, as follows:
“Reported anticancer effects include induction of apoptosis, ROS generation, cell-cycle arrest, inhibition of cell migration and invasion, loss of mitochondrial membrane potential, modulation of estrogen-related pathways, and antiangiogenic activity. In addition to these mechanistic findings, several isoflavones demonstrated significant tumor-growth inhibition in xenograft models, reinforcing their translational potential.”
3. Page 2, Line 68 – The statement “Particularly regarding its anticancer potential, as previously demonstrated in earlier studies” is not a complete sentence, please revise.
Reply: I removed the sentence that did not make sense and rewrote the text to make it more comprehensible, connecting one reference to the other:
“Recent studies have shown that, in the past ten years, approximately 1,036 new prenylated flavonoids have been isolated from 127 plant species, among which 219 are isoflavones [7]. This expands the prospects for investigating their biological activities. In a recently published review, newly obtained synthetic or hybridized derivatives were shown to exhibit strong antitumor activity [8].”
4. Table 1 requires significant reorganization to improve clarity and readability. Please reconsider the structure so that the information is easier for readers to interpret. Currently, the term “cytotoxicity” is used broadly; however, it simply refers to the property of being toxic to cells. To enhance the scientific value of the table, please revise the table to focus on the specific mechanisms by which these compounds exert cytotoxic effects. Additionally, verify all entries against the original sources, as several inconsistencies were noted. The following points represent a non-exhaustive list of issues observed:
Reply: We appreciate the reviewer’s comments regarding Table 1 and Figure 1. Table 1 follows the editorial standards previously accepted by the journal and will undergo professional formatting during the production stage, which typically enhances clarity, alignment, and visual consistency. However, we made a full revision of the Table and now we think that it´s clearer and more simplified.
We appreciate the reviewer’s suggestion regarding the refinement of cytotoxicity descriptors in Table 1. However, it is important to clarify that cytotoxicity assays constitute the foundational and most frequently reported evidence of antitumor activity in studies involving newly identified isoflavones. For many compounds, especially those recently isolated, the only available data consist of cell-viability assays such as MTT, CCK-8, SRB, or trypan blue exclusion, which quantify cell death without specifying the underlying molecular mechanism.
Because these assays determine IC₅₀ values, proliferation inhibition, and basal cytotoxic profiles across different cell lines, they provide essential information for: establishing potency, defining selectivity toward tumor vs. normal cells, guiding subsequent mechanistic studies, and allowing comparisons across compounds.
For this reason, cytotoxicity appears in the table only when it is the sole experimentally validated activity available from the primary article. We agree that the scientific value improves when mechanistic data are available, and therefore all studies reporting apoptosis, ROS generation, cell-cycle arrest, migration inhibition, or other pathways were classified under these specific mechanisms rather than under general cytotoxicity.
To address the reviewer’s concern, we revised the table by:
Explicitly distinguishing “non-mechanistic cytotoxicity assays” (“Cytotoxicity (MTT), no mechanistic data available”). In this way we hope to clarify this general term in Table 1.
4.1 Cytotoxicity is incorrectly spelled as ‘Citotoxicity’.
Reply: Corrected
4.2 Reference 23 – Please verify the IC₅₀ value listed (17 µM) against the original publication. The reported values in the source do not appear to match. Ensure accuracy and consistency with the cited reference.
Reply: The IC₅₀ value provided in the table was an approximate average of all the cell lines tested with the cyclodextrin–genistein complex. To make it clear we changed to the range of IC50 in all tumor cells (eg. from 1 – 10 uM).
4.3 Reference 35- The statement “The expression of the CYP24A1 gene induction is strongly amplified by hypoxia, the typical microenvironment of solid tumors” does not describe the anti-cancer mechanism of Genistein. Please revise.
Reply: You are correct, and we excluded this reference from the review since it didn´t elucidate any antitumoral effect of the substance. Since Formononetin remained with only one reference we had to move it to the supplementary table.
4.4 Reference 25 – “antitumor xenograft studies”. Antitumor xenograft studies represent an experimental approach, not a mechanism of action.
Reply: Corrected. Further We changed the Colum title from “Antitumor activity and/or mechanism” to “Antitumor mechanism and/or experimental evidence”
4.5 Reference 53 – “Tumor growth inhibition (Ehrlich carcinoma)”. Not an anticancer mechanism. Please revise.
Reply: We changed the Colum title from “Antitumor activity and/or mechanism” to “Antitumor mechanism and/or experimental evidence”
4.6 Please add a separate section highlighting compounds that have been evaluated in in-vivo studies. This will help readers distinguish between preclinical evidence and cell-based data.
Reply: The addition of a new section would make the table more complex and less clear. Further, in the table we describe the antitumor activity along with the mechanism of action (when available). We changed the Column title by adding the topic ‘experimental evidence’ in order to make it more comprehensive. Although a xenotumor is not, by itself, an anticancer activity or mechanism, tumor regression represents a preclinical piece of evidence. The same is valid for gene expression regulation. Additionally, as stated in the title, the scope of our work covers the mechanisms of action as well as the experimental evidence.
5. Page 9, Lines 117 and 122. The sentences “One possibility is that genistein may suppress Src-driven proliferative activity” and “Some derivatives also have great anticancer activity” are not scientifically rigorous. Please revise to provide clear, evidence-based statements and specify mechanisms or pathways supported by data.
Reply: We rewrote it to make it clearer:
“This compound suppresses the proliferative activity of tumor cells expressing constitutively active Src. In gallbladder carcinoma cells transfected with the v-src oncogene (HAG/src3-1), genistein significantly reduces cell growth compared with the corresponding control line transfected with the pSV2neo plasmid (HAG/neo3-5). These findings demonstrate that genistein effectively inhibits the proliferation of Src-transformed cells under the experimental conditions tested, without implying additional mechanistic pathways beyond those directly supported by the reported data.”
6. Page 9, Line 134 to 37 and Page 10, Line 183 – The sentences describing cyclodextrin and phytosome formulations of genistein would fit better in the paragraph at Line 138, which focuses on different formulations/delivery systems. In general, please restructure this section to consolidate all delivery systems into a single paragraph for improved flow and coherence. This will also eliminate redundant discussion of the same delivery system across multiple paragraphs in the current version.
Reply: We thoroughly rewrote this paragraph as suggested by the reviewer.
7. Page 14, Lines 352 and 354 – Please add a sentence explaining the difference between Wighteone, Lupiwighteone, and Isowighteone.
Reply: I added a section in the first paragraph that describes the differences:
“Wighteone typically carries a prenyl substituent at the C-6 position of the A-ring, whereas lupiwighteone features the prenyl group at C-8, making it a positional isomer. Isowighteone, as the name suggests, is an isomer of wighteone, differing by shifts in the position of hydroxyl and/or prenyl groups within the aromatic rings.”
8. The authors should consider adding a table that describes and distinguishes the natural and synthetic isoflavones discussed in the manuscript.
Reply: We thank you for the suggestion, but we consider that it is not necessary to make this modification to the table, because its purpose is to summarize the anticancer activity described in each article. In the text, we have already emphasized the natural source from which the molecules were isolated to allow this distinction between natural and synthetic.
9. Page 10, Line 155: The sentence “Genistein downregulates the expression of estrogen receptor… (Bax/BCL-2)” is unclear and scientifically incorrect. Please revise for clarity and accuracy.
Reply: We rewrote the passage to make the wording more scientifically appropriate:
“Genistein downregulates the expression of estrogen receptor ERα and phospho-ERα in MCF-7 cells, and modulates both mitochondria-independent and mitochondria-dependent apoptotic pathways through the activation of caspases 8 and 9. Moreover, genistein treatment enhances Bax gene expression while suppressing BCL-2 expression.”
10. Page 10, Line 171: The sentences “Using 3D culture models, it triggers apoptosis in a dose-dependent manner in PC-3 cells [13]’’ and “The downregulation of microRNA-155 (miR-155) and its target genes related to apoptosis and glucose metabolisms, such as STAT3 and hexokinase 2, particularly in tamoxifen-sensitive 174 MCF-7 cells [12]” are unclear and scientifically incorrect. Please revise for clarity and accuracy.
Reply: We rewrote the passage to make the wording more scientifically appropriate:
“This isoflavone induces dose-dependent apoptosis in PC-3 cells encapsulated in alginate hydrogel within a 3D culture model through non-mitochondrial pathway. Demonstrating that this culture system may enhance the effectiveness of treatment with this compound.[17]. Additionally, it downregulates microRNA-155 (miR-155) and its target genes involved in apoptosis and glucose metabolism, such as STAT3 and hexokinase 2, particularly in tamoxifen-sensitive MCF-7 cells [12].”
11. Page 13, Line 314: The sentence “The tumor volume was drastically reduced by ~90% with the combination of daidzein and carboplatin” is inaccurate. The reference paper uses cisplatin, and not carboplatin. These are two different compounds.
Reply: We rewrote it, correcting the cisplatin part:
“The tumor volume was drastically reduced by ~90% with the combination of daidzein and cisplatin,…”
12. Page 15, Line 392: Typo ‘coumpond’ instead of ‘compound’.
Reply: We rewrote it
“This compound also binds to the Aryl…”
Round 2
Reviewer 1 Report
Comments and Suggestions for Authors
Minor revision is necessary before this manuscript is recommended for acceptance.

The English could be improved to more clearly express the research.
Author Response
Point-by-point Rebuttal Letter
Reviewer 1:
1) “The organization of Table 1 is highly informative but could be improved for readability and precision. Currently, the "Antitumor mechanism and/or experimental evidence" column contains entries with varying levels of detail, some quite lengthy. This can make it difficult to quickly scan and compare mechanisms across different isoflavones. For instance, the entry for Biochanin A includes a long list of effects that might be better served by concise, standardized terminology. Could the authors consider restructuring this table? One suggestion would be to split this column into two: one for a concise, bolded primary mechanism (e.g., Apoptosis induction, Cell cycle arrest) and a subsequent column for additional experimental notes or specific pathways.”
Reply: We appreciate the reviewer’s constructive suggestion regarding the restructuring of Table 1. After careful consideration, we decided not to split the column “Antitumor mechanism and/or experimental evidence” for the following reasons:
- Only a small subset of compounds presented the “varying levels of detail” mentioned by the reviewer. These specific entries were originally longer simply because these compounds are more extensively studied and have a larger body of experimental evidence available in a given reference.
- Creating an additional column for “primary mechanism” would result in a highly asymmetric table, where only a few compounds would populate this new column while most cells would remain blank. We believe this would reduce clarity rather than improve it.
- Adding a new column would significantly compress the visual layout of the table, negatively affecting the readability of other key parameters already included.
In response to the reviewer’s concern, we revised and condensed the few overly long entries, ensuring they now match the level of detail provided for the other compounds. Only the compounds with genuinely broader experimental support remain slightly more detailed, reflecting the actual state of the literature.
We believe these adjustments improve readability while preserving the integrity and comparability of the information in Table 1.
1.2) “Furthermore, some IC50 values are presented as ranges (e.g., "18.6-34.3 μM") while others are single values. Please verify and standardize the presentation of all quantitative data for consistency.”
Reply: We thank the reviewer for this observation. We have reported the IC₅₀ values for all tested cell lines in µM, and, when applicable, we also included the corresponding percentage of inhibition, as previously requested. This dual presentation ensures completeness and accuracy; however, it also naturally increases the complexity of the table. Nonetheless, we have verified the numerical data and standardized the format to the extent possible while maintaining scientific precision.
1.3) “...in the Abstract, the phrase "Data were searched in the PubMed scientific literature database" could be refined to "A literature search was conducted using the PubMed database" for a more active and standard academic tone. In Section 3.1, the sentence "Genistein is the most studied soy-derived isoflavone. Among the 83 articles referenced in this review, 20 address the antitumor activity of genistein, accounting for approximately 17% of the published literature." The second sentence is somewhat redundant after the first. Consider combining them: "Genistein is the most extensively studied soy-derived isoflavone, with 20 out of the 83 articles referenced in this review (approximately 24%) addressing its antitumor activity."”
Reply: Thank you for the suggestion. We corrected.
1.4) “Additionally, there are instances of minor punctuation issues and hyphenation (e.g., "cell death" sometimes appears as "cell-death"). A thorough review to ensure consistency in terminology and style throughout the manuscript is recommended.”
Reply: Thank you for the suggestion. We corrected.
2) “The manuscript would benefit from a careful proofreading to address minor grammatical inconsistencies and improve the flow of certain sentences.”
For example, in the Abstract, the phrase "Data were searched in the PubMed scientific literature database" could be refined to "A literature search was conducted using the PubMed database" for a more active and standard academic tone. In Section 3.1, the sentence "Genistein is the most studied soy-derived isoflavone. Among the 83 articles referenced in this review, 20 address the antitumor activity of genistein, accounting for approximately 17% of the published literature." The second sentence is somewhat redundant after the first. Consider combining them: "Genistein is the most extensively studied soy-derived isoflavone, with 20 out of the 83 articles referenced in this review (approximately 24%) addressing its antitumor activity." Additionally, there are instances of minor punctuation issues and hyphenation (e.g., "cell death" sometimes appears as "cell-death"). A thorough review to ensure consistency in terminology and style throughout the manuscript is recommended.
Reply: We did a full and careful proofreading to address all minor grammatical inconsistencies. We hope that the manuscript is now ready for acceptance.
3) “The inclusion of a chemical structure figure is excellent. However, the caption for Figure 1, "Representation of the core structures of the isoflavones addressed throughout this review," is currently too vague.”
Reply: Thank you for the suggestion. We attempted to interpret the reviewer’s comment regarding the caption being “too vague.” To address this, we revised the caption to improve clarity and specificity. The updated version now reads:
“Figure 1. Schematic representation of the fundamental chemical scaffolds and substituent patterns that characterize the isoflavones examined throughout this review.”
We hope this revised wording aligns with the reviewer’s expectations for greater precision.